# The ALINFA Intervention Improves Diet Quality and Nutritional Status in Children 6 to 12 Years Old

**DOI:** 10.3390/nu15102375

**Published:** 2023-05-18

**Authors:** Naroa Andueza, Nerea Martin-Calvo, Santiago Navas-Carretero, Marta Cuervo

**Affiliations:** 1Department of Nutrition, Food Sciences and Physiology, Faculty of Pharmacy and Nutrition, University of Navarra, 31008 Pamplona, Spain; nandueza@unav.es (N.A.);; 2Center for Nutrition Research, University of Navarra, 31008 Pamplona, Spain; 3Department of Preventive Medicine and Public Health, School of Medicine, University of Navarra, University Campus, 31008 Pamplona, Spain; nmartincalvo@unav.es; 4Biomedical Research Networking Center for Physiopathology of Obesity and Nutrition (CIBERObn), Institute of Health Carlos III, 28029 Madrid, Spain; 5Navarra Institute for Health Research (IdiSNA), 31008 Pamplona, Spain

**Keywords:** diet quality, children, nutritional intervention, Kidmed index

## Abstract

The study aimed to evaluate the efficacy of a new nutritional intervention, focused on improving the quality of the diet in children aged 6 to 12 years. A 2-month parallel, controlled randomized trial was conducted in the Spanish child population. The children were randomized to ALINFA nutritional intervention, which consisted of a normocaloric diet that incorporates products, ready-to-eat meals and healthy recipes specifically designed for the study, or a control group, which received the usual advice on healthy eating. The change in diet quality was assessed through the Kidmed index. The secondary outcomes were anthropometry, glucose and lipid profiles, inflammation markers, dietary intake and lifestyle. The participants in the intervention group showed an increase in the mean score of the Kidmed index (*p* < 0.001). Alongside that, these children decreased their intake of calories (*p* = 0.046), and total and saturated fat (*p* = 0.016//*p* = 0.011), and increased fiber intake (*p* < 0.001). Likewise, the children in the ALINFA group increased the intake of white fish (*p* = 0.001), pulses (*p* = 0.004), whole grains (*p* < 0.001) and nuts (*p* < 0.001), and decreased fatty meat (*p* = 0.014), refined grain (*p* = 0.008), pastry (*p* < 0.001), fast food (*p* < 0.001) and sugar (*p* = 0.001) intake. Moreover, these children had a significantly decreased BMI (*p* < 0.001), BMI z-score (*p* < 0.001), waist circumference (*p* = 0.016) and fat mass (*p* = 0.011), as well as leptin (*p* = 0.004). Participants in the control group did not report significant changes in diet quality. In conclusion, ALINFA nutritional intervention is possibly a useful strategy to increase the diet quality in children, which is associated to improvements in the nutritional status. These results highlight the importance of developing well-designed nutritional interventions.

## 1. Introduction

Childhood is an important life stage for nutritional wellbeing, as eating habits developed during that time tend to track throughout adolescence and adulthood [1,2]. The consolidation of healthy dietary habits at an early age may prevent the development of future chronic diseases [3,4,5]. In addition, poor dietary habits are associated with the presence of overweight and obesity, as well as their comorbidities [6]. Many institutions, including the World Health Organization, are warning about the alarming increase in the rates of excess body weight across many population groups, with children being among the most worrying [7]. The probability of suffering food-related diseases in the adult life increases if they are set at an early age [3].

Dietary habits are influenced by multiple factors, including demographic, personal and environmental factors and food preferences, among others [5,6,8]. Food preferences, which are also defined at an early age [3], are a strong predictor of present and future food intake, and therefore the diet quality [5,8]. In this context, the availability and repeated exposure to healthy foods is the key to developing preferences and overcoming food dislikes [6,8,9]. For example, introducing a variety of fruits and vegetables and limiting exposure to secondary foods (those non-essential foods that are characterized by a high energy density, and by providing large amounts of saturated fat, salt and/or sugar) from an early age have been found to be important strategies for improving the later diet quality [8].

Globalization and urbanization lead to important changes in lifestyle and food patterns, with an increase in the consumption of highly processed foods with low nutritional density [10,11,12]. As a consequence, most children of western populations do not achieve the dietary goals of nutritional guidelines [12,13]. Spanish children are not an exception [14,15]. A recent study pointed out that ultra-processed foods have replaced fresh foods in the dietary pattern in Spanish households [14].

The Mediterranean diet is widely recognized as a healthy eating pattern, which has consistently demonstrated its favorable effects on health and the overall quality of life [16,17,18,19]. Research has shown that there is a strong inverse relationship between adherence to the Mediterranean dietary pattern and body mass index (BMI) in children and adolescents [20,21,22]. Moreover, apart from the positive effect on BMI, it has been observed that adherence to an intensive lifestyle intervention based on the recommendations of the Mediterranean diet allows achieving an optimal quality diet in children and adolescents [20,23]. Therefore, modifying the eating pattern toward the characteristic Mediterranean diet, can be an effective strategy to achieve a healthy diet and prevent health problems traditionally associated with low-quality dietary patterns [24,25].

The evidence on nutritional interventions aimed at improving children’s diet quality is limited and inconsistent, so it has not been possible to reach an agreement about the minimum characteristics that interventions must have to be successful [26,27,28]. In addition, there are hardly any studies in the general child population (regardless of their body weight) evaluating the effect that improving the quality of the diet may have on more than one health/nutritional variable such as anthropometric variables, body composition and/or serum glucose and lipid profiles, among others. Nutritional interventions of this type have been developed mainly in obese population [23,29,30]. Additionally, in the last decade, many interventions have been carried out in schools, especially in the cafeterias of the school canteens. Although these studies have shown that they can be effective, it is not fully determined if these school-based interventions result in changed behavior outside the school setting [31,32,33].

Given that eating habits begin to develop at an early age [34], the school stage, and specifically the period of primary education (6 to 12 years), should be the period on which future nutritional interventions focus.

In this scenario, the ALINFA project (in Spanish: healthy, accessible and affordable food for children) was developed to evaluate the efficacy of a new nutritional intervention focused on improving diet quality of children aged 6 to 12 years. In addition, we aimed to analyze the changes in anthropometry and body composition, biochemical parameters and lifestyle factors (physical activity, quality of life and eating behavior) associated with the intervention.

## 2. Materials and Methods

### 2.1. Study Design and Participants

The ALINFA study was a 2-month parallel, controlled randomized trial carried out at the Center for Nutrition Research of the University of Navarra in two periods in 2021 (March to June, and September to November).

This intervention study was the last part developed within a larger collaboration project of the same name (ALINFA), of which the objective was to provide healthy, affordable and accessible food products, in order to facilitate a healthier diet within children at a school age. The first stages of the project mainly focused on the design and development of all the products, ready-to-eat meals (detailed information in Appendix A) and healthy recipes that were intended to be included in the intervention study. Thus, the ALINFA diet was designed, which is based on the recommendations of the Mediterranean diet, with the novelty that it includes different products, ready-to-eat meals and healthy recipes developed specifically for children, with the aim of seeking alternatives that increase the adherence to the consumption of all healthy food groups. Three research centers (National Center for Technology and Food Safety -CNTA-, Public University of Navarre -UPNA- and University of Navarra -UNAV-) and five companies from the food sector (GRUPO APEX (Aperitivos y Extrusionados, S.A.), Irigoyen Comedor Saludable S.L., Harivenasa S.L., IAN S.A.U.: Navarra Food Industries, and Alimentos Sanygran S.L.), all located in Navarra, Spain, participated in this project.

The study was aimed at Spanish children aged 6 to 12 years old; therefore, it was intended to obtain a representative sample of this population group.

To be included in the study, participants had to be between 6 to 12 years old and have lunch at home or not eat at the school canteen. The exclusion criteria were: functional or structural anomalies of the digestive system; uncontrolled endocrine disorders; previous cancer in the last 5 years; weight loss/gain of ≥3 kg in the last three months; food allergies or intolerances; cognitive and/or mental impairment.

This study was designed in accordance with the Good Clinical Practice as stated by the Declaration of Helsinki [35], and ethical approval was obtained from the Research Ethics Committee of the University of Navarra (ref. 2021.027). The parents of all participants signed an informed consent form prior to the inclusion in the study. The study was registered in the ClinicalTrials.gov database (NCT05249166).

Participant recruitment was performed at schools (by sending fact sheets to guardians), through internal communication channels of the participating centers and local media. Randomization was stratified by sex and age, with a 2:1 ratio for the ALINFA and control group, respectively. Given that the control group would receive only instructions on a healthy diet and was comparable to standard community protocols, the researchers did not consider a 1:1 allocation ratio necessary. In the case that siblings participated in the study, they were assigned to the same group to facilitate for the family’s compliance with the intervention. In that case, the older sibling was randomized.

### 2.2. Description of the Intervention

In both groups, ALINFA and the control, the recommendations were based on a Mediterranean dietary pattern. In addition, the ALINFA group received an intensive intervention based on following a diet within a predesigned fixed full-day meal plan designed by dietitian–nutritionists from the University of Navarra.

The fixed full-day meal plan consisted of a prescribed diet including products, ready-to-eat meals and healthy recipes designed to provide at least 50% of energy from carbohydrates, less than 35% of energy from fat and 15–20% of energy from protein. The prepared dishes and products were provided to children and their siblings (even if they were not participating in the study), while the recipes were explained and provided to families to be made at home, so they had to be made at home. More specifically, the intervention consisted of a 2-week menu which contained 5 meals (this way the families repeated the prescribed diet 4 times throughout the study), with information on culinary techniques, ingredients and food quantities.

The ALINFA diet included ready-to-eat dishes (produced by Irigoyen Comedor Saludable S.L.), healthy recipes to develop at home (designed by CNTA) and food products developed by the study companies as part of the project, within the framework of a balanced diet. The products and dishes developed within the ALINFA consortium were the following: (a) Two mid-morning snacks based on corn and peas or fruit (GRUPO APEX: Aperitivos y Extrusionados S.A.), (b) three types of oatmeal toppings to add to yoghurts (Harivenasa S.L.), (c) three flexitarian products based on vegetable extrusion and meat (meat balls, nuggets and hamburgers developed by Alimentos Sanygran S.L.), (d) three precooked dishes (oatmeal paella, vegetable stew and vegetables with pasta developed by IAN S.A.U.: Navarra Food Industries). The meals prepared by Irigoyen Comedor Saludable S.L. were frozen to recook in the microwave, and they consisted of 10 dishes to cover 5 lunches, and 5 dishes to cover 5 dinners during a 2-week period. In addition, CNTA collaborated with a chef who specifically designed 15 recipes for the project, covering both main meals (lunch and dinner) and snacks, since two of the recipes were healthy sweets. With all these prepared dishes, recipes and healthier products, the research aimed at encouraging children to consume healthy food groups in different and original ways, including those foods that are generally not liked by the child population, such as fish or vegetables. To achieve this objective, the ready-to-eat meals and food company products were minimally processed foods, without additives, with a reduced salt content and adjusted to the nutritional needs of children (Appendix A).

In order to adjust to the nutritional requirements of all participants, two menus were prepared with different portion sizes depending on the age group (6–9 years and 10–12 years).

Given that breakfasts or mid-morning are not offered at Spanish schools and the children who participated in the study did not eat lunch in the school canteen, this facilitated compliance with the prescribed diet, since during school hours, they were not exposed to other foods and the families were in charge of preparing the food that they ate mid-morning.

The control group received the usual nutritional advice according to the Spanish Society of Community Nutrition food pyramid, putting special emphasis on portion size [36]. There was no energy restriction in either of the two nutritional interventions.

Another component of the intervention was nutritional education. This was carried out during the visits with the dietitian–nutritionist (at the Center for Nutrition Research of the University of Navarra), and was aimed at both the participating children and their parents. The control group received general dietary recommendations, while in the ALINFA group, a complete nutritional plan was explained. Through this nutritional plan, the children and their families were able to learn what a balanced diet consists of, the food groups that should be consumed preferably and their frequencies, as well as the portion that they should consume of each food for their age. In addition, the ALINFA prescribed nutritional plan-incorporated foods that are not generally consumed by this population. For example, some cereals such as oats, vegetables such as beets or cauliflower, and fish such as hake. In this way, the participating children had the opportunity to try new foods and new ways of consuming them, thus being able to modulate their food preferences.

The study was divided into 4 visits: V_0_: study information and screening; V_1_: start of the intervention (day 0); V_2_: follow-up visit (day 28); V_3_: end of the intervention (day 56). At V_1_ and V_3_, participants and their parents completed diet and lifestyle questionnaires. In those visits, anthropometric and body composition measures, as well as blood samples were collected. Participants in the ALINFA group received the food products and the consumption compliance record at the start and follow-up visit. The follow-up visit consisted of an interview to assess the follow-up of the intervention. Due to the COVID-19 situation, this visit was held online in the control group, but face-to-face in the intervention group for the food supply.

### 2.3. Measurements

#### 2.3.1. Dietary Assessment

In order to evaluate the diet quality, the Kidmed index was applied. This index is calculated from a validated questionnaire to assess the adherence to the Mediterranean dietary pattern in children and adolescents [37]. The Kidmed index assesses the adequacy of the Mediterranean dietary pattern in children through 16 items with a yes/no response. The maximum score that can be obtained is 12 points. This score is useful to classify the diet quality into three categories based on the score obtained: low diet quality (≤3 points), need to improve dietary pattern (4 to 7 points) and optimal Mediterranean diet (≥8 points).

Alongside that, dietary information was collected through a 138-item semi-quantitative food frequency questionnaire (FFQ) of the SENDO project, validated in children [38]. For each item, a portion size adapted to the child population was specified. Parents reported pre and post intervention how often their child had consumed each food. The pre-intervention data referred to food consumption during the previous year and the post-intervention data to the 2-month period that the study lasted. The nutrient content was calculated using data from the Spanish food composition database [39].

In addition, the participants of the ALINFA group completed a food record, in which they filled out the proportion of food consumed of the prescribed diet.

#### 2.3.2. Anthropometric, Blood Pressure and Body Composition

All measurements were determined under fasting conditions. Height was measured using a wall-mounted stadiometer (Seca 220, Vogel and Halke, Hamburg, Germany). Weight and body composition (fat, lean and muscular mass) were determined via bioimpedance (SC-330, Tanita, Tokyo, Japan). The BMI was calculated using the standard formula, weight (kg)/height (m)^2^. The BMI z-score was calculated and interpreted using the classification proposed by the World Health Organization (WHO) [40]. Waist circumference was determined using a flexible and inelastic anthropometric tape, while, for blood pressure, an automatic sphygmomanometer (IntelliSense. M6, OMRON Healthcare, Hoofddorp, The Netherlands) was used.

#### 2.3.3. Biochemical Parameters

Blood samples were drawn under fasting conditions. To obtain serum and plasma, the samples were processed in a standard centrifuge (Eppendorf 5804R, Hamburg, Germany). Serum glucose, total cholesterol (TC) and high-density lipoprotein cholesterol (HDL-c) levels were measured using colorimetric methods with a Pentra C200 autoanalyzer (Horiba ABX Diagnostics, Montpellier, France). Insulin, tumor necrosis factor (TNF-α), leptin, C-reactive protein (CRP) and interleukin 6 (IL-6) concentrations were quantified in an automated ELISA processing system DSX^®^ Dynex Technologies (Palex, Chantilly, VA, USA). Low-density lipoprotein cholesterol (LDL-c) concentration was calculated using the Friedewald formula [41].

#### 2.3.4. Other Questionnaires

In addition to the Kidmed index and the FFQ, the participants and/or their parents filled in other questionnaires.

The KINDL, which is an acronym for “child quality of life questionnaire” in German, is a validated self-administered questionnaire to evaluate the quality of life in children and adolescents [42,43]. It consists of 12 questions, and the responses are collected on a 3-point Likert-type scale. The highest scores represent a better self-perceived quality of life. All items were converted into a score ranging on a scale from 0 to 100 points.

The Physical Activity Questionnaire for Children (PAQ-C) is a validated self-administered questionnaire designed to assess children’s physical activity [44,45]. It consists of 10 items. The overall result of the questionnaire may range between 1 and 5, with the higher score indicating a higher level of physical activity.

The Child Eating Behavior Questionnaire (CEBQ) is a validated questionnaire aimed to assess children’s eating behavior [46,47]. It consists of 35 questions divided into eight eating behaviors: food responsiveness, enjoyment of food, emotional overeating, desire to drink, satiety responsiveness, slowness in eating, emotional undereating, and food fussiness. This questionnaire is completed by parents, and the answers are collected using a 5-point Likert-type scale.

### 2.4. Statistical Analysis

The sample size calculation was made a priori for a difference in change in the primary outcome (change in the Kidmed index score) of 1.5 points (sd = 1.5), with a power of 90%, assuming a bilateral alpha risk of 0.05 and for an intervention–control ratio of 2:1. We determined that 40 children were needed (27 in the intervention group and 13 in the control group). Considering a dropout rate of 30%, the total sample needed was estimated to be 52 (34 in the intervention group, and 18 in the control group). Continuous variables were described as the mean standard deviation, and the categorical variables as percentages. The normality of the data was evaluated using the Shapiro–Wilk test. Within-group differences were evaluated using a paired Student’s *t*-test or Wilcoxon test for quantitative variables, with a McNemar test or Chi-square test for the trends in categorical data. Between-groups differences were evaluated using Student’s *t*-test or the Mann–Whitney U test for quantitative variables, with a Chi-square test for categorical data. Given the multiple secondary outcomes, the Benjamini–Hochberg procedure was applied to these variables to correct for multiple testing [48]. Multiple linear regression models were performed to identify those dietary factors that could predict the change in some of the secondary variables. To do this, generalized estimating equation models were used to calculate the magnitude of the association of each predictor after accounting for the intra-cluster correlation between siblings.

The analyses were carried out with STATA 15.1 (Statacorp LP, College Station, TX, USA). The graphs were carried out with Microsoft Excel (Microsoft Office 2019 Professional Plus).

## 3. Results

The participant flow chart is shown in Figure 1. Initially, 101 children were interested in participating in the study, of which 10 did not meet the inclusion criteria collected in the pre-screening via telephone, 20 decided not to participate and 2 did not meet the inclusion criteria at the screening visit. Finally, 69 children took part in the study and were randomly assigned to the ALINFA group (*n* = 47) or the control group (*n* = 22), following a 2:1 ratio. Of these 69 children, 55 completed the intervention, with 44 children belonging to the ALINFA group and 11 the control group. When the dropouts were analyzed, 3 were due to non-compliance with the protocol and 11 due to loss of contact. This fact shows large differences in the dropout rate depending on the assigned group: 6.4% versus 50% (*p* < 0.001) in the control and ALINFA groups, respectively (Figure 1). The baseline characteristics of the participants who dropped out versus those who completed the intervention were analyzed and no significant differences were found (Appendix A).

Therefore, the final sample consisted of 55 participants (54.5% girls, aged 9.07 ± 1.73 years). Table 1 shows the baseline characteristics of the study population. No significant differences were observed between the groups at baseline. The mean score of the Kidmed index at baseline in the overall population was 7.03 ± 1.97 points, indicating the need for improvement of the dietary pattern, and after the intervention it increased within the ALINFA group, whereas it was unchanged in the control group, resulting in significant differences between the groups (*p* = 0.024). After the intervention, the ALINFA group achieved an optimal quality dietary pattern (Table 2). The adherence to ALINFA diet collected through food records determined that the percentage of the mean adherence was 75.36 ± 11.8.

Furthermore, the ALINFA group showed a significant increase in height (*p* < 0.001), and a decrease in BMI (*p* < 0.001) and the BMI z-score (*p* < 0.001). In the between-group comparisons, no significant differences were observed, although there was a trend toward statistical significance in the case of the BMI. On the other hand, the ALINFA group showed a significant decrease in leptin levels (*p* = 0.004), although the between-group comparison did not reach statistical significance. No differences were observed in the control group in any of the variables studied.

Lifestyle factors (physical activity and quality of life) and eating behavior did not show significant changes.

Table 3 shows the proportion of children who answered the healthiest option to each of the items in the Kidmed index. After the intervention, a significant increase in the proportion of the ALINFA group children that had a second piece of fruit every day (*p* = 0.004) consumed fresh or cooked vegetables once a day (*p* = 0.025) or more than once a day (*p* = 0.012), at least two servings/week of fish (*p* = 0.022), cereals or grains for breakfast (*p* = 0.025) and at least two servings/week of nuts per week (*p* = 0.004). Likewise, a significant reduction in the proportion of children that went to a fast-food restaurant more than once/week (*p* = 0.028), and consumed commercially baked goods for breakfast (*p* = 0.003) was also evidenced. Despite these findings, no significant differences were observed between the groups across any of the parameters analyzed.

When analyzing nutrient intake, the control group reported a higher intake of sodium at baseline (Table 4). The ALINFA group showed a significant decrease in energy intake of −175 ± 461.9 on average (*p* = 0.046). Likewise, a decrease of −8.45 ± 18.98 g of total lipids, and of −3.71 ± 7.59 g of saturated lipids (*p* = 0.016/*p* = 0.011, respectively) was observed. A significant increase in fiber intake of 4.2 ± 7.03 g was also observed (*p* < 0.001). Nevertheless, no significant differences between the groups were observed after the intervention. Regarding food group consumption, the intake of the following groups were analyzed: whole diary, low-free fat dairy, egg, lean and fatty meat, white and fatty fish, vegetable, fruit, pulse, refined and whole grains, nut, olive oil, other fats, pastries/confectionery, fast food, sugars, sweetened foods, water and salt (Appendix A). No differences in the mean consumption were observed at baseline. The ALINFA group reported a significant increase in the consumption of white fish (*p* = 0.001), pulses (*p* = 0.004), whole grains (*p* < 0.011) and nuts (*p* < 0.001), and a significant decrease in the consumption of fatty meat (*p* = 0.014), refined grains (*p* = 0.008), pastries and confectionery (*p* < 0.001), fast food (*p* < 0.001) and sugars (*p* = 0.001). In the between-group comparison no significant differences were observed. However, a trend toward statistical significance was observed in the following groups: fatty meat, whole grains and fast food (Figure 2).

Linear regression models were constructed to investigate the possible associations between different dietary factors and secondary variables that changed after the intervention. Specifically, a model that could predict the BMI z-score based on the change in the consumption of food groups, adjusted for the change in caloric intake, was developed. In addition, the possible correlation between siblings was taken into account. Of all the food groups analyzed, a significant association was observed between the intake of fast food and pastries/confectionary, and the BMI z-score (Table 5).

## 4. Discussion

The results of the present study showed that the ALINFA nutritional intervention was effective for improving the diet quality in a sample of Spanish children aged 6 to 12 years. Furthermore, this intervention also resulted in favorable changes in anthropometry, body composition, and glucose and lipid metabolism markers. We observed a significant improvement in diet quality, as determined with the Kidmed index, which was confirmed through the changes in the FFQ. To our knowledge, this is one of the first randomized trials in a pediatric population that employs this index to assess the change in diet quality [20,23]. This improvement was mainly due to the increase in the consumption of fruits, vegetables, white fish, whole grains, nuts and dairy, and the decrease in the consumption of fatty meat, refined grains, pastries, fast food and sugars. Our results agree with previous studies that reported nutritional interventions aiming at improving diet quality generally lead to a greater intake of fruits and vegetables [31,32,49,50,51,52], and a lower consumption of juices and sugary drinks [31,33,51,53]. Given that it has been feasible to increase the consumption of healthy foods and reduce the consumption of those considered unhealthy, we conclude that the intervention was effective and that our results are encouraging.

All the dietary results observed in the ALINFA group show that after the intervention, the eating pattern of the participating children resembled the Mediterranean diet [54,55].

Parallel to the changes in the diet, the ALINFA group showed significant improvements in the macronutrient intake. The decrease in energy, and total and saturated lipid intake was probably associated with the reduction in the consumption of fatty meat, fast food and pastries [56,57]. On the other hand, increased fiber intake is probably associated with the increased consumption of fruits, vegetables and whole grains. Given the changes in food consumption, the improvement in the dietary profile is not surprising [56,57,58]; however, it is not very often reported. A recently published review showed that only 33% of intervention studies focused on improving the quality of children’s diets reported the nutrient distribution of the participants’ diet [59]. Comparing this nutrient intake with the Dietary Reference Intakes for the Spanish population of the Spanish Federation of Nutrition, Food and Dietetic Societies (FESNAD) [60], it is observed that in the case of total fats, there is no exact recommendation, and as for saturated fats, the recommendation is that the intake be as low as possible. Therefore, the objectives of the reference dietary guidelines are being met as their intake has decreased after the intervention. On the other hand, in terms of fiber intake, the recommendations for children 6 to 12 years old range between 25 and 31 g/day. Given the increase in their intake produced after the intervention, reaching an average intake of approximately 25 g/d, the objective set out in the guide is met.

Related to dietary outcomes, the fact that the participants of the ALINFA group filled out a record of consumption of the prescribed foods allows for confidence in the observed results, especially when the percentage of adhesion is high.

Previous evidence showed that improving the diet quality resulted in favorable changes in nutritional status, body weight and body composition in the child and adolescent population [23,61]. It is worth noting that in this study, the weight loss trend observed in the ALINFA group was at the expense of fat mass, maintaining lean and muscular mass. Regarding the biochemical parameters, the ALINFA group showed a decrease in serum levels of leptin. On the other hand, it has been previously published that nutritional and lifestyle interventions that promote weight loss are often accompanied by a reduction in leptin levels. Furthermore, it has been suggested that reduced leptin levels may influence the maintenance of weight loss in children [62,63].

The results of the regressions carried out have allowed for the identification of factors associated mainly with the change in the BMI z-score. These factors have been the change in the consumption of two food groups, i.e., fast food and pastries/confectionary.

The main difference of the ALINFA strategy compared to previous intervention studies is that all ready-to-eat dishes, healthy recipes and food products were specifically designed for this population group and the acceptability of these products in the said population had been previously evaluated. Alongside that, most food items were directly provided to families, which facilitates the adherence to the assigned nutritional plan. Although this is not the first study to provide food items to the participants, previous studies in children were restricted to one meal and mostly in the school environment [49,52,64]. In addition, in this study we provided food supplies to the participant and his/her siblings, which also facilitated the adherence to the assigned nutritional plan because the participant perceived that his/her whole family was involved in the dietary change. These two elements may explain part of the effectiveness of the intervention.

In this context, another component of this study to highlight is the collaboration with companies in the food industry. The important role that these play in the eating habits of the population is well known. For this reason, different authors have pointed out the need for collaboration between public agents and companies to promote and facilitate healthy and accessible food [65,66,67].

Nutritional education has turned out to be a useful tool in the acquisition of healthy eating habits and modification of food preferences in the child population [68,69,70,71]. In addition, in some studies it has even been observed that it can produce sustained favorable effects on the BMI z-score [70]. In the context of the ALINFA study, the intensive nutritional education received by the ALINFA group during the visits with the dietitian may have been another of the components responsible for the effectiveness of the intervention in this group.

Different authors have pointed out that trying to promote more than one healthy behavior at one time may have a synergistic effect, resulting in a higher probability of improving at least one of them [49,72,73]. However, we did not observe any significant changes in the KINDL, PAQ-C or CEBQ questionnaires. This study did not aim at increasing physical activity, which was highly limited due to the COVID situation. We were also unable to demonstrate that dietary improvements lead to a better quality of life and eating behaviors, as suggested by Vajdi et al. [74].

The difference in the dropout rate between the groups is noteworthy (6.4% in the ALINFA group vs. 50% in the control group). This difference suggests that having a prescribed nutritional plan and being supplied with the necessary foods helps overcome the efforts required to make dietary changes [75,76,77]. Therefore, it is possible that participation in this type of study may serve to motivate children and their families to adopt healthier lifestyles.

In short, the results of the study indicate that there are different factors that lead to an increase in the quality of the diet of the children, and on which future research and health policies could focus; in the first place, the fact that providing healthy food to children already makes them consume it or at least try it and get used to it. Although the provision of food to families on a continuous basis is not sustainable, it could be a useful strategy to be implemented in schools or in certain population groups at risk, in order to modulate the food preference toward these foods. Second, designing healthy foods specifically for this population can increase their consumption. In this sense, the food industry plays an important role, being able to modify and create new accessible and affordable products with an adequate nutritional profile. Third, the involvement of children from an early age in their diet leads to the direct improvement in their diet and the acquisition of good eating habits. It would be interesting if studies such as this could be applied to larger population groups and for longer periods.

Despite the results, this study has some limitations. First, we acknowledge the large number of losses during the follow-up, especially in the control group. Although the final number of participants was slightly below the calculated sample size in the control group, significant differences could be observed in the main outcome. However, we cannot completely discard the fact that the lack of significant results in other variables may be due to a lower statistical power, since this was lower than expected at the beginning of the study. Thus, this study could be considered a pilot study. Second, the duration of the intervention did not allow us to evaluate long-term efficacy. Lastly, some results are based on self-reported data, which may lead to a misclassification bias, although nowadays, it is the most widely used research method for dietary assessment [78,79].

On the other hand, this study has several strengths. First, it is one of the first randomized and controlled trials including Spanish children, that ensures food supply and analyzes not only dietary changes, but also other possible effects of the intervention, such as changes in anthropometry, body composition, biochemical parameters and lifestyle. Second, we used questionnaires validated for the pediatric population. Third, the main outcome was evaluated using two different tools (the Kidmed index and the FFQ), which helped show the reliability of our results.

## 5. Conclusions

The ALINFA nutritional intervention was a useful strategy to improve the diet quality of children aged 6 to 12 years. These improvements in dietary quality were associated with favorable changes in the nutritional status.

In this case, the fact that the foods included in the nutritional plan were specifically designed for this population may have been a key element, as well as the fact that the whole family was involved in the change.

However, studies with a larger sample size and a longer follow-up time are needed to confirm the effectiveness of this intervention. For future research, it would be interesting to assess the effect of this intervention together with a physical activity intervention.

## Figures and Tables

**Figure 1 nutrients-15-02375-f001:**
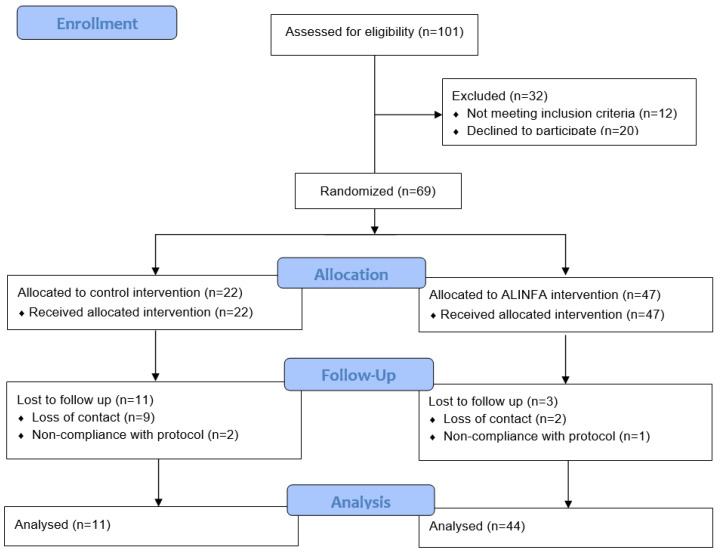
Allocation of the subjects of the ALINFA study according to the CONSORT 2010.

**Figure 2 nutrients-15-02375-f002:**
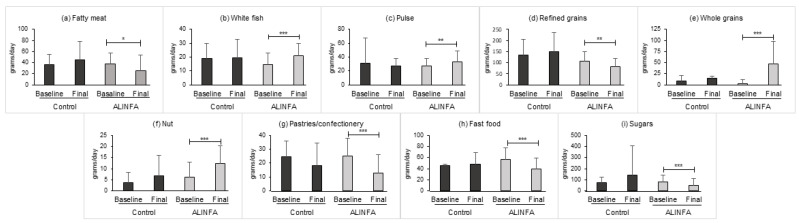
Statistically significant change after the intervention in the main food groups in the control (*n* = 11) and ALINFA groups (*n* = 44). Data are the mean ± SD. * *p* < 0.05; ** *p* < 0.01; *** *p* < 0.001. Benjamini–Hochberg adjustment was applied.

**Table 1 nutrients-15-02375-t001:** Baseline characteristics of the children included in the ALINFA study depending on the group.

	All (*n* = 55)	Control (*n* = 11)	Alinfa (*n* = 44)	*p*-Value ^a^
%	100%	20%	80%	
Gender (boys/girls)	25/30	7/4	18/26	0.176
Age (years)	9.07 ± 1.73	8.81 ± 1.53	9.12 ± 1.78	0.590
* Anthropometry *				
Weight (kg)	34.84 ± 8.34	35.23 ± 8.76	34.74 ± 8.33	0.863
Height (m)	1.38 ± 0.09	1.38 ± 0.09	1.38 ± 0.09	0.928
BMI (kg/m^2^)	17.88 ± 2.93	18.23 ± 3.21	17.80 ± 2.88	0.916
BMI z-score	0.09 ± 0.96	0.25 ± 0.93	0.05 ± 0.97	0.554
Waist (cm)	61.62 ± 7.51	62.01 ± 7.66	61.52 ± 7.55	0.924
SBP (mmHg)	101.9 ± 10.12	104.0 ± 10.75	101.4 ± 10.01	0.155
DBP (mmHg)	67.24 ± 9.03	67.86 ± 9.63	67.0 ± 8.98	0.802
* Body composition *				
Fat mass (kg)	7.93 ± 4.12	8.33 ± 4.42	7.82 ± 4.09	0.808
Lean mass (kg)	26.86 ± 4.84	26.9 ± 4.79	26.85 ± 4.91	0.975
Muscular mass (kg)	25.42 ± 4.60	25.46 ± 4.59	25.41 ± 4.66	0.975
Total water (kg)	20.05 ± 4.18	19.7 ± 3.50	20.14 ± 4.36	0.858
* Questionnaires *				
Quality of life (KINDL)	89.54 ± 4.56	89.14 ± 5.33	89.64 ± 4.41	0.803
Physical activity (PAQ-C)	3.08 ± 0.53	3.0 ± 0.63	3.10 ± 0.51	0.611
Eating behavior (CEBQ)				
- Food responsiveness	3.42 ± 0.76	3.81 ± 0.76	3.32 ± 0.88	0.100
- Enjoyment of food	2.21 ± 1.00	2.18 ± 1.07	2.22 ± 1.00	0.865
- Emotional overeating	1.83 ± 0.76	1.61 ± 0.76	1.89 ± 0.75	0.150
- Desire to drink	2.07 ± 0.89	1.78 ± 0.76	2.15 ± 0.89	0.183
- Satiety responsiveness	2.82 ± 0.38	2.87 ± 0.36	2.80 ± 0.39	0.631
- Slowness in eating	2.47 ± 0.48	2.29 ± 0.35	2.51 ± 0.50	0.179
- Emotional undereating	2.17 ± 0.73	1.95 ± 0.74	2.22 ± 0.73	0.277
- Food fussiness	2.92 ± 0.30	2.85 ± 0.33	2.93 ± 0.29	0.430
Diet quality (Kidmed index)				
Total punctuation	7.03 ± 1.97	6.90 ± 2.11	7.06 ± 1.95	0.813
Interpretation:				
- Low diet quality	4 (7.27%)	1 (9.10%)	3 (6.82%)	
- Need to improve dietary pattern	28 (50.91%)	5 (45.45%)	23 (52.27%)	0.911 ^b^
- Optimal MD	23 (41.82%)	5 (45.45%)	18 (40.91%)	
* Biochemistry *				
Glucose (mg/dL)	93.21 ± 5.44	93.84 ± 3.62	93.03 ± 5.89	0.698
Insulin (µIU/mL)	9.76 ± 4.03	8.88 ± 2.75	10.00 ± 4.33	0.467
Total cholesterol (mg/dL)	173.5 ± 23.46	175.7 ± 25.28	172.9 ± 23.31	0.752
HDL-c (mg/dL)	62.26 ± 9.44	64.94 ± 13.25	61.51 ± 8.18	0.341
LDL-c (mg/dL)	99.88 ± 21.51	99.89 ± 21.46	99.87 ± 21.91	0.998
TNF-α (pg/mL)	5.07 ± 0.98	4.94 ± 0.96	5.11 ± 1.00	0.642
Leptin (ng/mL)	1.72 ± 1.37	1.88 ± 1.23	1.68 ± 1.42	0.369
CRP (mg/dL)	0.87 ± 0.90	1.05 ± 1.24	0.82 ± 0.80	0.752
IL-6 (pg/mL)	24.26 ± 51.66	15.50 ± 41.46	26.72 ± 54.51	0.075

Abbreviations: BMI, body mass index; SBP, systolic blood pressure; DBP, diastolic blood pressure; PAQ-C, physical activity questionnaire for children; CEBQ, child eating behavior questionnaire; HDL-c, high-density lipoprotein cholesterol; LDL-c, low-density lipoprotein-cholesterol; TNF-α, tumor necrosis factor-alpha; CRP, C-reactive protein; IL-6; interleukin 6. Data are the mean ± SD. ^a^
*p*-values based on the Student’s *t*-test or Mann–Whitney U. ^b^ *p*-values based on the Chi-square test. Statistical significance defined as *p* < 0.05.

**Table 2 nutrients-15-02375-t002:** Changes produced after the intervention in the ALINFA study depending on the group, in anthropometry, body composition, diet quality, lifestyle and biochemical parameters.

	Control (*n* = 11)			ALINFA (*n* = 44)			Change between Groups (*p*-Value) ^b^
	Pre-Intervention	Post-Intervention	*p*-Value ^a^	Pre-Intervention	Post-Intervention	*p*-Value ^a^	
Gender (boys/girls)	7/4			18/26			
Age	8.81 ± 1.53			9.12 ± 1.78			
* Anthropometry *							
Weight (kg)	35.23 ± 8.76	35.94 ± 8.87	0.217	34.74 ± 8.33	34.44 ± 8.14	0.195	0.082
Height (m)	1.38 ± 0.09	1.39 ± 0.09	0.062	1.38 ± 0.09	1.39 ± 0.09	<0.001	0.772
BMI (kg/m^2^)	18.23 ± 3.21	18.34 ± 3.12	0.774	17.80 ± 2.88	17.36 ± 2.55	<0.001	0.082
BMI z-score	0.25 ± 0.93	0.21 ± 0.88	0.774	0.05 ± 0.97	−0.09 ± 0.83	<0.001	0.280
Waist (cm)	62.01 ± 7.66	61.87 ± 9.85	0.917	61.52 ± 7.55	60.19 ± 7.20	0.016	0.861
SBP (mmHg)	104.0 ± 10.75	103.5 ± 9.80	0.917	101.4 ± 10.01	103.1 ± 8.66	0.514	0.675
DBP (mmHg)	67.86 ± 9.63	67.40 ± 8.65	0.917	67.09 ± 8.98	67.43 ± 8.12	0.868	0.190
* Body composition *							
Fat mass (kg)	8.33 ± 4.42	8.15 ± 4.07	0.657	7.82 ± 4.09	7.29 ± 3.56	0.011	0.776
Lean mass (kg)	26.9 ± 4.79	27.79 ± 5.30	0.103	26.85 ± 4.91	27.00 ± 5.11	0.563	0.186
Muscular mass (kg)	25.46 ± 4.59	26.31 ± 5.06	0.093	25.41 ± 4.66	25.54 ± 4.85	0.577	0.186
Total water (kg)	19.7 ± 3.50	20.35 ± 3.87	0.111	20.14 ± 4.36	20.16 ± 4.62	0.556	0.206
* Questionnaires *							
Quality of life (KINDL)	89.14 ± 5.33	89.64 ± 3.94	0.881	89.64 ± 4.41	90.27 ± 3.95	0.577	0.776
Physical activity (PAQ-C)	3.0 ± 0.63	3.08 ± 0.56	0.849	3.10 ± 0.51	3.18 ± 0.48	0.514	0.822
Eating behavior (CEBQ)							
- Food responsiveness	3.81 ± 0.76	3.75 ± 0.65	0.774	3.32 ± 0.88	3.33 ± 0.90	0.938	0.776
- Enjoyment of food	2.18 ± 1.07	1.98 ± 0.64	0.774	2.22 ± 1.00	2.28 ± 0.91	0.474	0.675
- Emotional overeating	1.61 ± 0.76	1.79 ± 0.87	0.774	1.89 ± 0.75	1.90 ± 0.68	0.789	0.776
- Desire to drink	1.78 ± 0.76	1.94 ± 0.77	0.443	2.15 ± 0.89	1.99 ± 0.77	0.304	0.412
- Satiety responsiveness	2.87 ± 0.36	2.69 ± 0.37	0.521	2.80 ± 0.39	2.72 ± 0.35	0.389	0.776
- Slowness in eating	2.29 ± 0.35	2.20 ± 0.56	0.774	2.51 ± 0.50	2.49 ± 0.48	0.806	0.776
- Emotional undereating	1.95 ± 0.74	2.11 ± 0.99	0.774	2.22 ± 0.73	2.24 ± 0.76	0.904	0.776
- Food fussiness	2.85 ± 0.33	3.07 ± 0.40	0.111	2.93 ± 0.29	2.9 ± 0.25	0.564	0.097
Diet quality (Kidmed index)	6.90 ± 2.11	7.72 ± 0.68	0.081	7.06 ± 1.95	9.18 ± 1.55	<0.001	0.024
Interpretation:							
- Low diet quality	1 (9.10%)	0 (0%)		3 (6.82%)	1 (2.27%)		
- Need to improve dietary pattern	5 (45.45%)	5 (45.45%)	0.463 ^c^	23 (52.27%)	7 (15.91%)	<0.001 ^c^	
- Optimal MD	5 (45.45%)	6 (54.55%)		18 (40.91%)	36 (81.82%)		
* Biochemical parameters *							
Glucose (mg/dL)	93.84 ± 3.62	94.74 ± 5.22	0.800	93.03 ± 5.89	90.85 ± 6.81	0.248	0.776
Insulin (µIU/mL)	8.88 ± 2.75	12.55 ± 4.67	0.336	10.00 ± 4.33	11.09 ± 7.41	0.909	0.186
Total cholesterol (mg/dL)	175.77 ± 25.28	168.66 ± 18.93	0.750	172.93 ± 23.31	167.31 ± 20.63	0.248	0.776
HDL-c (mg/dL)	64.94 ± 13.25	63.68 ± 10.65	0.800	61.51 ± 8.18	62.36 ± 9.10	0.665	0.901
LDL-c (mg/dL)	99.89 ± 21.46	89.68 ± 17.11	0.443	99.87 ± 21.91	93.16 ± 20.29	0.118	0.776
TNF-α (pg/mL)	4.94 ± 0.96	5.38 ± 1.31	0.638	5.11 ± 1.00	5.06 ± 1.47	0.868	0.529
Leptin (ng/mL)	1.88 ± 1.23	2.35 ± 1.87	0.881	1.68 ± 1.42	0.94 ± 0.65	0.004	0.823
CRP (mg/dL)	1.05 ± 1.24	2.47 ± 4.23	0.750	0.82 ± 0.80	1.00 ± 2.03	0.370	0.412
IL-6 (pg/mL)	15.50 ± 41.46	14.46 ± 35.03	0.840	26.72 ± 54.51	18.31 ± 42.75	0.564	0.776

Abbreviations: BMI, body mass index; SBP, systolic blood pressure; DBP, diastolic blood pressure; PAQ-C, physical activity questionnaire for children; CEBQ, child eating behavior questionnaire; HDL-c, high-density lipoprotein cholesterol; LDL-c, low-density lipoprotein cholesterol; TNF-α, tumor necrosis factor-alpha; CRP, C-reactive protein; IL-6; interleukin 6. Data are the mean ± SD. Secondary variables were adjusted using the Benjamini–Hochberg procedure. ^a^
*p*-values based on the Student’s *t*-test or Wilcoxon test. Statistical significance defined as *p* < 0.05. ^b^ *p*-values based on the Student’s *t*-test or Mann–Whitney U. Statistical significance defined as *p* < 0.05. ^c^ *p*-values based on the Chi-square test for trend. Statistical significance defined as *p* < 0.05.

**Table 3 nutrients-15-02375-t003:** Change in the results of the Kidmed index questionnaire between the groups after the intervention in the ALINFA study.

	Control (*n* = 11)		ALINFA (*n* = 44)		Change between Groups (*p*-Value) ^a^
	Pre-Intervention	Post-Intervention	Pre-Intervention	Post-Intervention	
Takes a fruit or fruit juice every day (+1)	63.64%	81.82%	79.55%	90.91%	0.70
Has a second fruit every day (+1)	36.36%	54.55%	43.18%	68.18% *	0.63
Has fresh or cooked vegetables regularly once a day (+1)	90.91%	81.82%	77.27%	95.45% *	0.10
Has fresh or cooked vegetables more than once a day (+1)	54.55%	54.55%	43.18%	65.91% *	0.25
Consumes fish regularly (at least 2–3/week) (+1)	81.82%	63.63%	63.64%	86.36% *	0.504
Goes >1/ week to a fast-food restaurant (hamburger) (−1)	9.09%	18.18%	15.91%	2.33% *	0.66
Likes pulses and eats them >1/week (+1)	90.91%	81.82%	81.82%	95.45%	0.12
Consumes pasta or rice almost every day (5 or more per week) (+1)	36.36%	36.36%	6.82%	6.82%	0.37
Has cereals or grains (bread, pasta, rice, etc) for breakfast (+1)	90.91%	90.91%	75.00%	93.18% *	0.10
Consumes nuts regularly (at least 2–3/week) (+1)	18.18%	18.18%	40.91%	72.73% *	0.30
Uses olive oil at home (+1)	100%	100%	100%	100%	1.00
Skips breakfast (−1)	0%	0%	4.55%	4.55%	0.61
Has a dairy product for breakfast (yogurt, milk, etc) (+1)	100%	100%	97.73%	93.18%	0.28
Has commercially baked goods or pastries for breakfast (−1)	36.36%	18.18%	43.18%	6.82% *	0.16
Takes two yoghurts and/or some cheese (40 g) daily (+1)	18.18%	27.27%	52.27%	65.91%	0.31
Takes sweets and candy several times every day (−1)	0%	0%	2.27%	2.27%	0.61

The data represent the percentage of children in each group who answered affirmatively to each of the items of the questionnaire. Benjamini–Hochberg adjustment was applied. * Statistically significant differences after the intervention in each group. *p*-values based on the McNemar test. Statistical significance defined as *p* < 0.05. ^a^
*p*-values based on the Chi-square test. Statistical significance defined as *p* < 0.05.

**Table 4 nutrients-15-02375-t004:** Differences in the content of the main macro- and micronutrients after the intervention in the ALINFA study.

	Control (*n* = 11)			ALINFA (*n* = 44)			Change between Groups (*p*-Value) ^b^
	Pre-Intervention	Post-Intervention	*p*-Value ^a^	Pre-Intervention	Post-Intervention	*p*-Value ^a^	
Kcal	2326 ± 457.7	2272 ± 647.8	0.800	2157 ± 378.7	1981 ± 367.0	0.046	0.776
Total carbohydrates (g)	251.5 ± 57.17	253.4 ± 100.6	0.917	219.6 ± 41.75	209.7 ± 47.33	0.389	0.776
Total lipids (g)	106.9 ± 20.47	99.42 ± 20.06	0.709	95.85 ± 19.69	87.06 ± 17.59	0.016	0.776
- Saturated	27.85 ± 5.37	24.41 ± 5.03	0.437	26.22 ± 7.38	22.51 ± 7.62	0.011	0.901
-Monounsaturated	39.09 ± 7.89	40.48 ± 7.71	0.774	39.77 ± 10.24	36.08 ± 6.12	0.076	0.949
- Polyunsaturated	12.43 ± 2.68	12.66 ± 3.56	0.904	11.66 ± 2.66	11.34 ± 2.57	0.670	0.404
Total protein (g)	96.80 ± 27.94	91.10 ± 24.30	0.799	96.20 ± 31.73	89.63 ± 18.50	0.737	0.776
Fiber (g)	21.93 ± 8.67	24.73 ± 6.71	0.662	20.84 ± 5.85	25.05 ± 6.30	0.001	0.868
Cholesterol (mg)	261.22 ± 25.48	277.11 ± 57.13	0.75	288.41 ± 75.75	258.12 ± 81.81	0.121	0.776
Sodium (mg)	4166 ± 1197 ^#^	3760 ± 973	0.638	3195 ± 793 ^#^	3015 ± 814	0.562	0.412

Data are the mean ± SD. Benjamini–Hochberg adjustment was applied ^#^ Significant differences between the groups at baseline. ^a^
*p*-values based on the Student’s *t*-test or Wilcoxon test. Statistical significance defined as *p* < 0.05. ^b^ *p*-values based on the Student’s *t*-test or Mann–Whitney U. Statistical significance defined as *p* < 0.05.

**Table 5 nutrients-15-02375-t005:** Association between the BMI z-score and food group intake.

	BMI z-Score		
	β	95% CI	*p*-Value
Fast food	0.0003	−0.001–0.002	0.693
Pastries/confectionary	0.001	−0.002–0.004	0.557
Fast food X Pastries/confectionary	−0.0002	−0.0003–−0.0001	<0.001
Kcal	0.0001	−0.078–0.032	0.029

β represents changes in outcomes for the increasing number of units of the BMI z-score punctuation in the entire population.

## Data Availability

All data and material are available upon reasonable request to the corresponding author.

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
