# Peer review of "The ALINFA Intervention Improves Diet Quality and Nutritional Status in Children 6 to 12 Years Old"

_nutrients, 2023, doi:10.3390/nu15102375_

Round 1
Reviewer 1 Report
This paper is well-written and very interesting, but some point must be considered.
1) The sample lost in the control group is very significant. Did authors calculate the study power to evaluate if this comparison is possible?
2) The use of industrialized products as health pattern to children is very worrying, as literature has demonstrated. Thus, more details about the ingredients and additives that are included in this food preparations must be detailed and discussed.
3) The table 5 and figure 2 are similar, even the authors should choose one of them to present.
4) The results concerning change in BMI, and metabolic biomarkers are very relevant. Thus, the authors must be this information in the title. Moreover, new analyses can be performed to evaluate the association between these changes and intervention (e.g. regression models, or correlations).
Author Response
REVIEWER 1
Comments and Suggestions for Authors
This paper is well-written and very interesting, but some point must be considered.
1) The sample lost in the control group is very significant. Did authors calculate the study power to evaluate if this comparison is possible?
First of all, we would like to thank you the reviewer for the useful comments and suggestions.
The statistical power calculation was performed, and the power is slightly below the 80% (73%) that was aimed a priori. We have inserted a comment in line 553, of the revised manuscript.
2) The use of industrialized products as health pattern to children is very worrying, as literature has demonstrated. Thus, more details about the ingredients and additives that are included in this food preparations must be detailed and discussed.
The relationship between the consumption of ultra-processed foods and their negative effect on health has been widely studied and demonstrated. Taking this into account, the food industry has been identified as an important ally in promoting health, by modifying the composition of its products. This improvement would mean an increase in the supply of healthy products on the market.
Specifically, in this study the collaboration was with five companies in the food sector. These foods included in the ALINFA diet were specifically designed and developed for this study following nutritional standards. They were minimally processed foods, with a reduced salt content and without additives.
The ALINFA consortium arose as an opportunity to promote healthy eating habits from childhood. For that, the food companies developed healthy food products, considering the food requirements and preferences of children.
The ingredient list for these foods has now been added as Supplementary information 2.
In addition, the manuscript has been modified in the following lines:
- Lines 164-167: “To achieve this objective, the ready-to-eat meals and food company products were minimally processed foods, without additives, with a reduced salt content and adjusted to the nutritional needs of children (Supplementary file, information S1-S2 and table S1).”
- Lines 510-514: “In this context, another component of this study to highlight is the collaboration with companies in the food industry. The important role that these play in the eating habits of the population is well known. For this reason, different authors have pointed out the need for collaboration between public agents and companies to promote and facilitate healthy and accessible food.”
3) The table 5 and figure 2 are similar, even the authors should choose one of them to present.
Thanks for the comment. It is true that the table 5 and the figure 2 contain the same information. For this reason, we have decided to present only in the manuscript the figure 2 that is the most informative and to present the table as supplementary table S3.
4) The results concerning change in BMI, and metabolic biomarkers are very relevant. Thus, the authors must be this information in the title. Moreover, new analyses can be performed to evaluate the association between these changes and intervention (e.g. regression models, or correlations).
We agree with the reviewer the results observed in the BMI and in the biochemical markers are very interesting. For this reason, we have decided to modify the title of the article (The ALINFA intervention improves diet quality and nutritional status in children 6 to 12 years old). In addition, we have performed new statistical analyses. Specifically, we have developed a multiple linear regression model to predict BMI (table 5, revised manuscript).
Reviewer 2 Report
Thanks for your initiative to improve nutrition among school children in Spain, and documenting the efforts made by dietitians, nutritionist and companies to provide a diet based solution for preventing future development of chronic diseases . This initiative planned to come up with a measured solution, but it has some limitations and missing information, some of which the authors have also mentioned in the discussion section.
One major problem is that there was unusually high rate of drop out (50%) among the control group, meaning the study has lots its power. The sample for intervention to control participants was designed as 2:1 and it started with 47 and 22 participants for intervention and control, respectively. At the end of the study, the sample was 44:11 participants for intervention and control, meaning it ended with a 4:1 ratio for participants. It is highly likely that the participant characteristics for control group were different between the baseline and end of the study, which the authors should check and report (e.g., supplementary data).
This high drop-out rate is likely to not have preserved the randomisation and balance between groups. The strength of randomised controlled trial is that randomisation would make the groups (intervention and control) comparable and thereby the effect can be attributable to the intervention. When the randomisation is not preserved, the findings can not be confidently attributed to the intervention.
The authors need to perform sensitivity analysis to compare group characteristics over time. They also could impute missing data and check its effect on the analysis, or at least mention it as a limitation. Due to small sample and high drop-out, the study became more like a pilot study rather than a well-implemented RCT.
Another limitation is that the intervention was planned as a prescriptive approach with a very defined meal plans for each days of the week. The intervention relied on products and ready to eat meals specified for each day of the week, but did not collect any data on adherence to such diet or acceptability of such an intervention. Clearly, this is a disruption to what the participants usually would have cooked/eaten at home, if they were not participating to the intervention. In other parts of the world, studies have shown that there may be food sharing between household members when a specific meal or product is given to a particular children/ mother participating to an intervention but not the whole family. They mentioned that the inclusion of siblings may have improved adherence, but it seems they have not collected the data on adherence.
The authors need to provide more information about how much would these products cost, who currently paid for it and how they see this intervention would sustain (i.e., whether the families would need to buy such productions to change dietary habits) after the intervention phase out. The heavy reliance of products and ready meals (another product it seems) would mean the families would have to access them through purchase or free distribution, and different families may have different purchasing capacity to afford such a diet plan. The authors have mentioned that schools might have to include such food provision to make them accessible to students. It is also that the study period is very short so it is not clear whether such a plan can be implemented for long.
Another missing information is that the study selected participants who eat lunch at home, but we do not know what percentage of children in Spain/ locality would normally eat at home. It is expected that a large percentage would be normally eating at school canteen and therefore such an intervention could be irrelevant for significant number of children. It is also difficult to understand how this could have worked in Spain, as in other countries children would either bring food from home or would eat at canteen. It is not clear how is it feasible to implement such an intervention.
abstract, line 31-32: It mentioned that the ALINFA nutritional intervention is possibly a useful strategy to increase diet quality in children, which is associated to improvements on anthropometry, but anthropometry is a broad term and can mean so many things. I think the authors meant overweight and obesity. Increase in weight could be a good thing for a malnourished child, but a bad thing for obese child, so it needs to be specifically discussed in relation to the context.
In the introduction, line 43 -44 they mentioned overweight and obesity as a disease, but it is a condition that increase risk of chronic diseases. The overweight or obesity itself is not a disease.
Issues with methods:
1. The authors uses BMI and BMI Z-score, but BMI is recommended for use in adult population and BMI Z score is used for school-aged children. They also need to mention which reference for BMI Z-score they have used.
2. Sampling and loss-to-follow up: Whereas the study already started with a small sample size of 22 participants in the control group, there is 50% loss to follow up in this group. The study recruited from schools, but it did not mention how many schools were participating, whether the study sample was adjusted for school level or household level clustering as siblings were recruited as well. There would be correlation between samples within a household. The study description of analysis did not provide information whether this correlation was considered in the analysis.
3. The authors have not done any comparison of characteristics of control group sample for baseline and end of study, as it is likely that the group characteristics would be very different.
4. This large decline in sample means the study lost power to detect difference between group. The essence of randomised controlled trial is that it aims to achieve baseline balance, and then at post-study the convention is to compare between group difference. However, the authors have presented a lot of results on within group changes over time, which dilutes the focus. Those results can be provided in supplementary file, rather than in the main body of the paper.
Design issue: The WHO recommendation of food and nutrition policy for schools in Europe (Implementing policies to restrict food marketing: a review of contextual factors (who.int); Microsoft Word - Food Nutrition Schools NFS FINAL.doc (who.int) suggests whole school approach, marketing restriction, and involving parents in the design of intervention, whereas this study proposed an intervention which is highly unlikely to sustain beyond the intervention implementation. This study seems to be biased towards introducing food products and strict diet plans among children rather than enabling them to make the right food choices within their means. The information about cost and availability of these products beyond the study is missing.
The authors should include the suggested guidance by WHO and UNICEF for dealing with overweight and obesity in European context and discuss the study within that context.
Assessment issues:
Dietary intake: Line 188-190: Dietary assessment is complicated when doing in-person, whereas this study did online data collection for control group's dietary intake and for the same did in-person data collection for intervention group, which would influence the data quality between the two groups.
Anthropometry and blood tests: It is also not clear why the participants were measured for fasting condition (both blood test and anthropometry were done at fasting condition, appears from the description), as it would influence their measurement. I do not see any reference suggesting that children of school age should be measured at fasting condition. The authors need to justify if this is a standard practice in this age group and it is recommended by some assessment protocol.
Sampling: I am not very clear about the study sampling, and why a 1.5 points change in mean KIDMED index was taken. To me, it seems that the study would have made a better decision if they rather aimed for an improvement in percentage of children having optimal dietary behaviour (based on KIDMED index definition). The authors did not mention what would be the medical/ public health significance of a 1.5 point change in KIDMED index, if it was based on some previous study findings/ expert consultations/ arbitrary.
Analysis: I suggest the authors read the publication and include some comment on multiplicity issues, as they have done many many tests. It would be a good idea to remain focused on the key primary and secondary outcomes.
Guowei Li, Monica Taljaard, Edwin R Van den Heuvel, Mitchell AH Levine, Deborah J Cook, George A Wells, Philip J Devereaux, Lehana Thabane, An introduction to multiplicity issues in clinical trials: the what, why, when and how, International Journal of Epidemiology, Volume 46, Issue 2, April 2017, Pages 746–755, https://doi.org/10.1093/ije/dyw320
I think a revised manuscript with further clarity and incorporating the limitations better can improve the manuscript, which can also make it more informative and useful for the readers.
The quality of English is generally acceptable.
Author Response
REVIEWER 2
Comments and Suggestions for Authors
Thanks for your initiative to improve nutrition among school children in Spain, and documenting the efforts made by dietitians, nutritionist and companies to provide a diet-based solution for preventing future development of chronic diseases. This initiative planned to come up with a measured solution, but it has some limitations and missing information, some of which the authors have also mentioned in the discussion section.
-One major problem is that there was unusually high rate of drop out (50%) among the control group, meaning the study has lost its power. The sample for intervention to control participants was designed as 2:1 and it started with 47 and 22 participants for intervention and control, respectively. At the end of the study, the sample was 44:11 participants for intervention and control, meaning it ended with a 4:1 ratio for participants. It is highly likely that the participant characteristics for control group were different between the baseline and end of the study, which the authors should check and report (e.g., supplementary data).
This high drop-out rate is likely to not have preserved the randomization and balance between groups. The strength of randomized controlled trial is that randomization would make the groups (intervention and control) comparable and thereby the effect can be attributable to the intervention. When the randomization is not preserved, the findings can not be confidently attributed to the intervention.
Thank you for your revision and comments; we are sure that the quality of the manuscript is now more solid. At the end of the study and after observing the high dropout rate, mainly in the control group, we decided to check if there was any baseline difference between the participants who dropped out and those who did not that could justify the dropout rate. The analyses determined that there were no baseline differences between the completers and those who dropped out. Therefore, the results observed in the ALINFA group can be attributed to the intervention.
This information has been added in the supplementary file (table S2).
The manuscript has been modified in the following lines:
- Lines 292-294: “Baseline characteristics of the participants who dropped out versus those who completed the intervention were analyzed and no significant difference were found (Supplementary File, table S2).”
-The authors need to perform sensitivity analysis to compare group characteristics over time. They also could impute missing data and check its effect on the analysis, or at least mention it as a limitation. Due to small sample and high drop-out, the study became more like a pilot study rather than a well-implemented RCT.
Within the different types of sensitivity analysis, one of the ways to deal with missing data is the complete case analysis method (Therese D et al., 2001). For this reason, in this study only the data of those participants who completed the intervention were analyzed.
Although, given the high dropout rate, the study could be considered as a first pilot study. It has been included in the revised manuscript as one of the limitations of the study (Lines 554: “So, this study could be considered as a first pilot study”).
Therese D. Pigott (2001) A Review of Methods for Missing Data, Educational Research and Evaluation, 7:4, 353-383, DOI: 10.1076/edre.7.4.353.8937
-Another limitation is that the intervention was planned as a prescriptive approach with a very defined meal plans for each days of the week. The intervention relied on products and ready to eat meals specified for each day of the week, but did not collect any data on adherence to such diet or acceptability of such an intervention. Clearly, this is a disruption to what the participants usually would have cooked/eaten at home, if they were not participating to the intervention. In other parts of the world, studies have shown that there may be food sharing between household members when a specific meal or product is given to a particular children/ mother participating to an intervention but not the whole family. They mentioned that the inclusion of siblings may have improved adherence, but it seems they have not collected the data on adherence.
Diet adherence is very interesting information. For this reason, the participants in the ALINFA group had to complete a dietary record throughout the study in which they filled in the amount of food consumed from the prescribed diet.
Since the ALINFA diet products and recipes were specifically designed for children, it was determined that it was sufficient to provide these foods to siblings and not to the entire family. In the case of the recipes that families made at home, they were encouraged to cook them for all family members.
Furthermore, the acceptability of these ready-to-eat dishes, food products and recipes were evaluated in small pilot studies carried out in each company. More studies are needed that focus on the evaluation of the acceptability and cost of these meals.
The manuscript has been modified in:
- Lines 217-218: “In addition, the participants of the ALINFA group completed a food record, in which they filled out the proportion of food consumed of the prescribed diet.”
- Lines 480-482: “Related to dietary outcomes, the fact that the participants of the ALINFA group filled out a record of consumption of the prescribed foods allows confidence in the observed results.”
-The authors need to provide more information about how much would these products cost, who currently paid for it and how they see this intervention would sustain (i.e., whether the families would need to buy such productions to change dietary habits) after the intervention phase out. The heavy reliance of products and ready meals (another product it seems) would mean the families would have to access them through purchase or free distribution, and different families may have different purchasing capacity to afford such a diet plan. The authors have mentioned that schools might have to include such food provision to make them accessible to students. It is also that the study period is very short so it is not clear whether such a plan can be implemented for long.
First, this intervention study is the last part of a strategic project of the same name funded by the Government of Navarra, Spain. Participating research centers received full funding and participating companies received at least half the funding needed to encourage them to develop these new healthy foods and products.
On the other hand, we understand that following this diet completely in the long term is not sustainable in many cases, but it is a first step in the development of these healthy products so that their offer is even greater. In addition, it has been useful for children to try new foods or new way of processing and cooking different foods, which may be more attractive for them, within a framework of healthy eating. In addition, given that part of the child population eats in school canteens, the possibility of implementing this intervention in schools is suggested to evaluate its effectiveness in this environment. In neither of these two intervention scenarios would it be feasible to include and consume all these foods continuously, but they can serve as a first step or transition towards a healthy diet. In fact, since both the ready-to-eat products and the products provided were minimally processed foods, most of them could be prepared and cooked at home with little or no recipe modification.
Finally, as indicated in one of the previous points, more studies are needed to evaluate the acceptability of these products. Although the results and follow-up of the intervention by the ALINFA group give an idea of their acceptability.
-Another missing information is that the study selected participants who eat lunch at home, but we do not know what percentage of children in Spain/ locality would normally eat at home. It is expected that a large percentage would be normally eating at school canteen and therefore such an intervention could be irrelevant for significant number of children. It is also difficult to understand how this could have worked in Spain, as in other countries children would either bring food from home or would eat at canteen. It is not clear how is it feasible to implement such an intervention.
On the page of the Ministry of Education and Professional Training of the Government of Spain it can be consulted the data of the children who use the school canteen in both public and private centers in the different educational stages (https://www.educacionyfp.gob.es/servicios-al-ciudadano/estadisticas/no-universitaria/centros/centrosyunid.html)
Specifically, the percentage of primary school students who used this service in the 2020/2021 academic year was 35.4% in Spain and 29.3% in Navarra (region in which the study was carried on). This very low rate is probably due to the COVID-19 situation, given that many schools began journeys of only seasons in the morning and dispensed with the school canteen.
However, if we look at the data from previous years, in the 2019/2020 academic year in Spain only 40.2% and in Navarra 52.6% of primary school students attended the school canteen. In the 2018/2019 academic year the figures were similar to those of the following academic year, in Spain a total of 38.9% of primary school students used the service and in Navarra 52.5%.
Therefore, although the percentage of children who go to the school canteen is still considerable, today, many children eat at home, so interventions that focus on them are necessary. In addition, given that in many schools children bring their own food from home, a nutritional intervention like this one, is feasible to implement since the children would simply have to bring the products that we provide to consume there.
-abstract, line 31-32: It mentioned that the ALINFA nutritional intervention is possibly a useful strategy to increase diet quality in children, which is associated to improvements on anthropometry, but anthropometry is a broad term and can mean so many things. I think the authors meant overweight and obesity. Increase in weight could be a good thing for a malnourished child, but a bad thing for obese child, so it needs to be specifically discussed in relation to the context.
Thank you for your comment. Indeed, even if we agree with you that the term anthropometry can be broad, in our study we did not focus on obesity or overweight. The main objective was to improve dietary quality and as a secondary objective to examine the impact that this improvement had on other variables related to the state of health, such as anthropometric variables in this case. Nevertheless, and to avoid confusion, the text has been modified rephrasing this term for nutritional status (line 31-33 revised manuscript: “In conclusion, ALINFA nutritional intervention is possibly a useful strategy to increase diet quality in children, which is associated to improvements on nutritional status”)
-In the introduction, line 43 -44 they mentioned overweight and obesity as a disease, but it is a condition that increase risk of chronic diseases. The overweight or obesity itself is not a disease.
We are sorry to disagree with you. Although it is true that overweight is not a disease, many institutions and organizations such as the World Obesity Federation, Centers for Disease Control and Prevention and the American Medical Association among others have already defined obesity as a disease.
The manuscript has been modified by removing the word disease (Lines 42-45: “Many institutions, including the world health organization, are warning about the alarming increase in the rates of excess body weight across all population groups, with children being one of the most worrying”).
Issues with methods:
- The authors uses BMI and BMI Z-score, but BMI is recommended for use in adult population and BMI Z score is used for school-aged children. They also need to mention which reference for BMI Z-score they have used.
Thank for your comment. The reference to the BMI z-score has been added to the manuscript in the material and methods section (Line 226-227: BMI z-score was calculated and interpreted using the classification proposed by the World Health Organization (WHO))
- Sampling and loss-to-follow up: Whereas the study already started with a small sample size of 22 participants in the control group, there is 50% loss to follow up in this group. The study recruited from schools, but it did not mention how many schools were participating, whether the study sample was adjusted for school level or household level clustering as siblings were recruited as well. There would be correlation between samples within a household. The study description of analysis did not provide information whether this correlation was considered in the analysis.
The recruitment of volunteers was carried out through different channels. In the case of schools, those responsible were provided with information about the study and sent the information to families. Recruitment in the schools themselves was not carried out as such.
In the new regression analyzes of relationships carried out, the possible correlation between siblings has been considered and controlled. For this, generalized estimation equations (GEE) were used.
This information has been clarified in the text:
-Lines 125-126: " Participant’s recruitment was performed at schools (by sending fact sheets to responsibles)"
- The authors have not done any comparison of characteristics of control group sample for baseline and end of study, as it is likely that the group characteristics would be very different.
A comparison of the baseline characteristics of the volunteers who dropped out vs those who completed the intervention was made, demonstrating that there were no differences in the characteristics measured at baseline between both groups (Supplementary File, table S2).
- This large decline in sample means the study lost power to detect difference between group. The essence of randomized controlled trial is that it aims to achieve baseline balance, and then at post-study the convention is to compare between group difference. However, the authors have presented a lot of results on within group changes over time, which dilutes the focus. Those results can be provided in supplementary file, rather than in the main body of the paper.
We agree with your comment. It is possible that there is an excess of information of results. For this reason, we have decided to transfer Table 5 to the supplementary materials (Supplementary File, Table S3).
Design issue:
The WHO recommendation of food and nutrition policy for schools in Europe (Implementing policies to restrict food marketing: a review of contextual factors (who.int); Microsoft Word - Food Nutrition Schools NFS FINAL.doc (who.int) suggests whole school approach, marketing restriction, and involving parents in the design of intervention, whereas this study proposed an intervention which is highly unlikely to sustain beyond the intervention implementation. This study seems to be biased towards introducing food products and strict diet plans among children rather than enabling them to make the right food choices within their means. The information about cost and availability of these products beyond the study is missing.
The authors should include the suggested guidance by WHO and UNICEF for dealing with overweight and obesity in European context and discuss the study within that context.
This study has not been carried out in schools, rather it is an intervention carried out in homes. We understand that in the long term, maintaining a strict diet based on the inclusion of these products is unfeasible. This intervention can serve to introduce new foods, new ways of consuming them, and modify children's food preferences towards a healthier profile. On the other hand, collaboration with companies in the food sector is aimed at encouraging them to develop new products or healthy foods.
These conclusions are commented on in the discussion section on the lines 535-547.
Assessment issues:
Dietary intake: Line 188-190: Dietary assessment is complicated when doing in-person, whereas this study did online data collection for control group's dietary intake and for the same did in-person data collection for intervention group, which would influence the data quality between the two groups.
In both cases, the collection of dietary information was carried out in person. This was collected at both the baseline and end of intervention visits, in which they had to attend to the facilities of the Center for Nutrition Research. The only visit that was carried out differently was the follow-up visit, which in the case of the control group was performed online due to the COVID-19 situation. In the case of the ALINFA group, it could not be done online since they had to collect the food to continue with the intervention. In this visit as such, no dietary information was collected, but an interview was conducted to assess the follow-up of the intervention.
The text has been modified to clarify it: "The follow-up visit consisted of an interview to assess the follow-up of the intervention" (lines 196-197, revised manuscript)
Anthropometry and blood tests: It is also not clear why the participants were measured for fasting condition (both blood test and anthropometry were done at fasting condition, appears from the description), as it would influence their measurement. I do not see any reference suggesting that children of school age should be measured at fasting condition. The authors need to justify if this is a standard practice in this age group and it is recommended by some assessment protocol.
By protocol, all measurements must be taken under the same conditions so that they can be compared over time. In this context the authors consider performing the measurements at fasting because it is the most uniform way to collect these measurements under the same conditions.
Sampling: I am not very clear about the study sampling, and why 1.5 points change in mean KIDMED index was taken. To me, it seems that the study would have made a better decision if they rather aimed for an improvement in percentage of children having optimal dietary behavior (based on KIDMED index definition). The authors did not mention what would be the medical/ public health significance of a 1.5 point change in KIDMED index, if it was based on some previous study findings/ expert consultations/ arbitrary.
This small change of 1.5 points we believed could be relevant as it would allow the change from one category of diet classification to another. In addition, it was considered that the majority of the population would have a medium quality diet.
At baseline, the volunteers' mean KIDMED score was 7.03 points. Based on this score, the diet is classified into three categories: low quality of the diet (≤ 3 points), need to improve the dietary pattern (4 to 7 points) and optimal Mediterranean diet (≥ 8 points). In this case, an increase of 1.5 points would allow a change to the higher classification, thus achieving a diet of optimum quality. Therefore, we believe that this change may be clinically relevant.
These same data can be interpreted as percentages. In the ALINFA group (group that improves their diet after the intervention) it is observed that the percentage of children who need to improve their diet decreases from approximately 40 to 15%, as the same time as the percentage of children who have a optimum quality diet increases from 40% to 80% approximately.
Analysis: I suggest the authors read the publication and include some comment on multiplicity issues, as they have done many many tests. It would be a good idea to remain focused on the key primary and secondary outcomes.
Guowei Li, Monica Taljaard, Edwin R Van den Heuvel, Mitchell AH Levine, Deborah J Cook, George A Wells, Philip J Devereaux, Lehana Thabane, An introduction to multiplicity issues in clinical trials: the what, why, when and how, International Journal of Epidemiology, Volume 46, Issue 2, April 2017, Pages 746–755, https://doi.org/10.1093/ije/dyw320
I think a revised manuscript with further clarity and incorporating the limitations better can improve the manuscript, which can also make it more informative and useful for the readers.
Thanks for the suggestion. Given the number of multiple comparisons made, we have decided to apply the Benjamini-Hochberg adjustment in secondary outcomes.
It has been clarified in the statistical analysis section that this procedure has been used in the lines 273-275, of the revised manuscript, “Given the multiple secondary outcomes, the Benjamini-Hochberg procedure to correct for multiple testing was applied on these variables [38].”
Reviewer 3 Report
The manuscript " The ALINFA study: a randomized controlled trial of a nutrition intervention improves diet quality in children 6 to 12 years old” presented to me for review is undoubtedly an interesting and very comprehensive approach to the topic of proper nutrition for children.
However, a few things could improve this manuscript.
Below are some suggestions to improve the manuscript.
[90] What is the main purpose of the work? Is it known that all these children required nutritional intervention.
[268] Why the 2:1 ratio was chosen. Was it a coincidence or was this the design of the study? In total, n=11 (4:1) were finally qualified. This is a bit small, so I would suggest modifying the title and adding pilot studies due to the small group.
Were all children within the normal BMI or within the percentile curves? I suggest that obese and underweight children should be excluded from the study, because the group is too small and this makes conclusions difficult.
With such a small sample size, the group should be as homogeneous as possible.
If the group was homogeneous, it should be added to the manuscript.
[492] What do anthropometric parameters mean? What parameters are important for this study? Was the idea to slim these children down?
Conclusions are not presented precisely.
Author Response
REVIEWER 3
The manuscript " The ALINFA study: a randomized controlled trial of a nutrition intervention improves diet quality in children 6 to 12 years old” presented to me for review is undoubtedly an interesting and very comprehensive approach to the topic of proper nutrition for children.
However, a few things could improve this manuscript.
Below are some suggestions to improve the manuscript.
[90] What is the main purpose of the work? Is it known that all these children required nutritional intervention.
The main objective of the study is the improvement of the quality of the diet. Furthermore, as secondary objectives, we wanted to study the possible changes that could result from this change in diet quality, such as anthropometry, body composition, lifestyle, and eating behavior.
This study is aimed at the child population in general, which could benefit from an intervention on diet at different levels: diet improvement, modulation of food preferences, increase in the variety of foods consumed and improvements in nutritional status, among others.
[268] Why the 2:1 ratio was chosen. Was it a coincidence or was this the design of the study? In total, n=11 (4:1) were finally qualified. This is a bit small, so I would suggest modifying the title and adding pilot studies due to the small group.
Given that the control group would receive only instructions, on healthy diet, and was comparable to standard community protocols, the researchers did not consider necessary a 1:1 allocation ratio. This paragraph has been added to the revised manuscript (Lines 128-130)
In addition, it is true that given the high dropout rate, this study could be considered more of a pilot study. However, after making a comparison of the baseline characteristics of those volunteers who dropped out versus those who completed the intervention, no significant differences were observed between groups. Therefore, no factor could be found that would have conditioned this high dropout rate (Supplementary file, table S2).
Thank you for the suggestion made about considering this study as a pilot study. For this reason, we have decided to add it as a limitation of the study (Line 554, revised manuscript), and now it stands: “So, this study could be considered as a first pilot”.
Were all children within the normal BMI or within the percentile curves? I suggest that obese and underweight children should be excluded from the study, because the group is too small and this makes conclusions difficult. With such a small sample size, the group should be as homogeneous as possible. If the group was homogeneous, it should be added to the manuscript.
The study was aimed at the general population, the idea is that it was a representative sample of this population group. Almost all of the children who participated in the study had a normal weight, according to the BMI z-score values. Although, we believe that it is not necessary to eliminate the data of children who were overweight or obese, since then it would not be a representative sample.
The text has been modified in the following lines:
- Lines 113-114: “The study was aimed at the general child population, it was intended to obtain a representative sample of this population group”.
[492] What do anthropometric parameters mean? What parameters are important for this study? Was the idea to slim these children down?
Thank you so much for the comments. It is true that the term anthropometry might be too general. For this reason, we have decided to replace this term with nutritional status, which we believe may be more appropriate.
The text has been modified in lines 568-570 " These improvements in dietary quality were in turn associated with favorable changes in nutritional status”-
The objective of the study was not weight loss, since the focus of the study was not overweight or obesity. This intervention focused on dietary improvement and, linked to this, the improvement of variables related to health status, including weight.
This result shows us that the intervention is not effective in a single measure, but also produces a positive effect on nutritional status, demonstrating the usefulness and effectiveness of the intervention.
Conclusions are not presented precisely.
We have modified the conclusions also attending to the previous suggestions. We hope that now they are clearer and represent a reliable information of the results.
Lines 567-577: “The ALINFA nutritional intervention was a useful strategy to improve the diet quality of children aged 6 to 12 years. These improvements in dietary quality were in turn associated with favorable changes in nutritional status. In this case, the fact that the foods included in the nutritional plan were specifically designed for this population target may have been a key element, as well as the fact that the whole family was involved in the change. However, studies with a larger sample size and longer follow-up time are needed to confirm the effectiveness of this intervention. For future research, it would be interesting to assess the effect of this intervention together with a physical activity intervention”.